# Allosteric Antagonism of the Pregnane X Receptor (PXR): Current-State-of-the-Art and Prediction of Novel Allosteric Sites

**DOI:** 10.3390/cells11192974

**Published:** 2022-09-24

**Authors:** Rajamanikkam Kamaraj, Martin Drastik, Jana Maixnerova, Petr Pavek

**Affiliations:** 1Department of Pharmacology and Toxicology, Faculty of Pharmacy, Charles University in Prague, Heyrovskeho 1203, 50005 Hradec Kralove, Czech Republic; 2Department of Physical Chemistry and Biophysics, Faculty of Pharmacy, Charles University in Prague, Heyrovskeho 1203, 50005 Hradec Kralove, Czech Republic

**Keywords:** PXR, pregnane X receptor, allosteric site, AF-2 site, BF-3 site, PAM-antagonist, CAR

## Abstract

The pregnane X receptor (PXR, *NR1I2*) is a xenobiotic-activated transcription factor with high levels of expression in the liver. It not only plays a key role in drug metabolism and elimination, but also promotes tumor growth, drug resistance, and metabolic diseases. It has been proposed as a therapeutic target for type II diabetes, metabolic syndrome, and inflammatory bowel disease, and PXR antagonists have recently been considered as a therapy for colon cancer. There are currently no PXR antagonists that can be used in a clinical setting. Nevertheless, due to the large and complex ligand-binding pocket (LBP) of the PXR, it is challenging to discover PXR antagonists at the orthosteric site. Alternative ligand binding sites of the PXR have also been proposed and are currently being studied. Recently, the AF-2 allosteric binding site of the PXR has been identified, with several compounds modulating the site discovered. Herein, we aimed to summarize our current knowledge of allosteric modulation of the PXR as well as our attempt to unlock novel allosteric sites. We describe the novel binding function 3 (BF-3) site of PXR, which is also common for other nuclear receptors. In addition, we also mention a novel allosteric site III based on in silico prediction. The identified allosteric sites of the PXR provide new insights into the development of safe and efficient allosteric modulators of the PXR receptor. We therefore propose that novel PXR allosteric sites might be promising targets for treating chronic metabolic diseases and some cancers.

## 1. Introduction

The pregnane X receptor (PXR, *NR1I2*) is a nuclear receptor superfamily member. Nuclear receptors are mostly activated by exogenous and endogenous compounds. Interestingly, the classified nuclear receptors conserve the DNA binding domain (DBD), but not its ligand-binding domain (LBD). PXR belongs to the *NR1I* subfamily together with the vitamin D receptor (VDR, *NR1I1*) and the constitutive androstane receptor (CAR, *NR1I3*) [1,2].

The PXR protein consists of an *N*-terminal conserved DBD and activation function domain 1 (AF-1), and the C-terminal contains an LBD with activation function 2 (AF-2). A flexible hinge region intervenes between the DBD and the LBD (Figure 1A). The artificial intelligence application AlphaFold has been used to predict the full length of the PXR structure; moreover, the error plot shows the expected position error for each residue in the sequence (Figure 1B) [3,4,5]. The PXR ligand-binding pocket (LBP) is formed by ~28 amino acids, which are typically hydrophobic residues. In the absence of an agonist, PXR can be found in the cytosol. A ligand-activated PXR forms a heterodimer with retinoid X receptor alpha (RXRα), and the complex binds coactivators (e.g., steroid receptor coactivator-1, SRC-1) (Figure 1C) [6,7,8].

The PXR is expressed in the liver, small intestine, and colon, and is traceable in the brain [9,10,11,12,13,14,15,16]. It functions as the master regulator of drug detoxification in the liver [17]. In a growing number of reports, ligand-activated PXRs alter the metabolic profile of a drug and increase the probability of drug–drug interactions (DDIs) [18,19,20].

Activation of PXRs causes significant clinically relevant DDIs with compounds whose clearance is critically dependent on hepatic biotransformation by inducible cytochrome P450s, such as *CYP3A4* (Figure 1D). Under this scenario, the metabolism of the drug is significantly augmented by a PXR ligand (inducer, perpetrator) resulting in decrease in therapeutic efficacy, which may subsequently require shortening dosage intervals or increasing dosage. The most serious DDI interactions mediated by PXR activation have been reported for rifampicin [21,22,23,24].

Activated PXRs play a role in many metabolic pathways, including bile acid metabolism, lipids, and glucose homeostasis, as well as in inflammation [25]. Recently, their potential role in tumor growth, aggressiveness, relapse, and cancer drug resistance was also noted in mouse tumor xenografts for colon cancer [26,27,28,29,30]. These various physiological functions make PXRs a potentially valuable therapeutic target [25,31]. Numerous synthetic or natural ligands, mostly with agonist activities, have been discovered among currently used therapeutics and dietary or natural compounds [32,33]. Activation of PXRs using their ligands has been connected with many metabolic side effects (e.g., hypercholesterolemia or liver steatosis) as well as with intestinal cancer [10,25,26,34,35,36]. Therefore, PXR antagonists could have a significant therapeutic value, although designing a novel PXR antagonist is challenging due to the promiscuous nature of the PXR LBP [37,38,39,40,41,42].

This review summarizes and discusses our current knowledge about the allosteric modulation of PXRs. In addition, we describe our discovery of the binding function 3 (BF-3) of PXRs, which is the common allosteric binding site for other nuclear receptors. We also mention the novel allosteric site III based on in silico modeling. We propose that knowledge of allosteric modulation of PXRs as well as the characterization of the novel allosteric binding III and BF-3 sites will help us understand the biology of the PXR as well as discover novel efficient PXR antagonists.

## 2. The Orthosteric Ligand-Binding Pocket of the PXR

The PXR shares common structural characteristics with other nuclear receptors [43]. The orthosteric binding site of the PXR receptor is large (>1600 Å^3^), dynamic, and flexible enough to bind bulky ligands [40]. The PXR LBD is characterized by an alpha-helical sandwich structure with a specialized five-stranded beta sheet [44]. The LBP is deeply embedded in the PXR LBD (Figure 2A) [45]. The orthosteric PXR LBP is formed by the helices α3/5/6/7/10/12 (Figure 2B) [46]. The PXR ligand-binding cavity was detected at the bottom of the LBD, with the ligand entrance located between the alpha 2 and 6 helices (Figure 2B) [40,44]. The PXR was observed as a homodimer in solution, and the crystal structure showed linking with β1′ strands and with support by six intermolecular hydrogen bonds in the monomer [43].

The PXR LBD shares structural similarities with the VDR and CAR, with a sequence homology of 49.4% and 48.6%, respectively. In humans, the main isoform of the PXR (NM_003889.4, variant 1) consists of 434 amino acids, and the main hydrophobic hot spots are the amino acid residues F288, W299, and Y306, which interact with all reported co-crystallized PXR ligands [40]. All available 49 PXR LBD crystal structures represent the PXR receptor in its active state and are available for coactivator interactions through helix 12 (H12) for the following transcriptional activation. Due to the destabilization of the PXR LBD, to date no crystal structure of a PXR antagonist has been reported. Therefore, as an alternative to crystallography, the biophysical technique of hydrogen-deuterium exchange mass spectrometry (HDX-MS) was used to analyze the PXR–antagonist interactions [41].

The structural characteristics of PXR ligands are determined by the nature of the PXR binding pocket: (i) the PXR LBP is large and highly hydrophobic; and (ii) the polar residues Ser247, Gln285, His407, and Arg410 define the ligand binding position in the PXR LBP via strong hydrogen bond (H-bond) interactions. Ser247 and Gln285 residues are involved in the orientation of the ligand binding and the His407 and Arg410 side chains are flexible to accommodate various sizes of ligands in the LBP [47].

Recently, a detailed examination of the PXR antagonist/inverse agonist SPA70 binding has been conducted. The distinction of the PXR antagonist/inverse agonist (SPA70) from agonist (SJB7) binding showed that the agonist should have the hydrogen bond to the polar residue of Ser247 in helix 10 as well as proximity to the helix 12 region. In contrast, the antagonist has interaction with Ser247 residue and/or has weak contacts with helix 12 [39]. Since the PXR agonist SJB7 and the antagonist SPA70 bind to the same ligand-binding pocket in the PXR in the combined agonist and antagonist conformation, it is difficult to discriminate agonist and antagonist binding sites in the PXR LBD [39,41]. Notably, a single PXR LBP mutation (W299A) converts the activity of SPA70 from inhibition to activation [48].

It has recently been reported that the designed PXR molecular glue SJPYT-195, which is composed of SPA70 linked with the CRBN ligand (thalidomide), degraded the GSPT1 translation termination factor instead of the PXR [49], but the loss of GSPT1 decreased the level of PXR protein in human colon cancer cells (SNU-C4) [49]. SJPYT-195 weakly bound the PXR LBD, suggesting that long-length linkers may be more favorable in the design of potent PROTACs for PXR [49,50]. We can also speculate that PXR allosteric sites could be another strategy for the successful targeting of PXR protein degradation (Figure 2) [51].

**Figure 2 cells-11-02974-f002:**
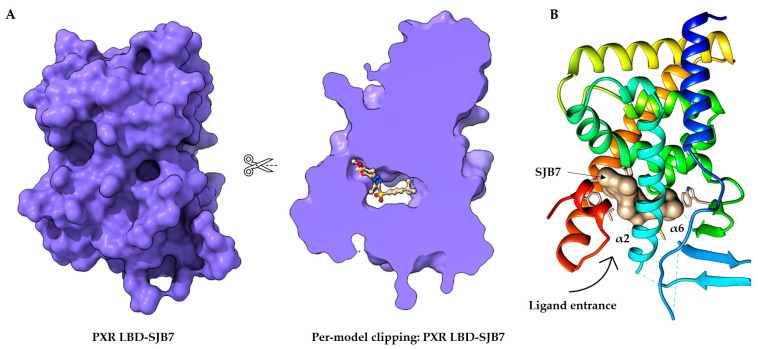
Structure of the orthosteric PXR ligand binding site. (**A**) Per-model clipping of the PXR ligand binding domain (to visualize invisible portions of a model). The PXR in PDB entry 5X0R has an SJB7 agonist ligand in an interior pocket, visualized using UCSF ChimeraX [41,52]. (**B**) The crystal structure of the PXR LBD complexed with SJB7. Cartoon representation of PXR- SJB7 binding in the orthosteric site. The PXR agonist SJB7 is indicated in brown, with the arrow indicating the entrance of the ligand. All structures are shown in the same orientation [41,44].

## 3. Allosteric Modulation of Nuclear Receptors

Allosteric modulators are usually structurally different from orthosteric ligands, and they bind to distinct sites that are spatially distant from orthosteric sites to modulate the activities of orthosteric ligands [53].

The essential features of a receptor allosteric modulation are: (1) orthosteric and allosteric sites not overlapping, i.e., there is no mutual biomolecular interaction in the binding; (2) the allosteric binding of one ligand to its site can affect the binding of the second ligand to the orthosteric site and vice versa; and (3) the effect of allosteric modulation can be positive or negative depending on the existing orthosteric ligand. This phenomenon is known as “probe dependence” [54,55]. Allosteric modulation may be inhibited due to protein–protein interactions (e.g., nuclear receptor–coactivator) or a co-bound receptor, such as in the case of the nuclear receptor homodimerization and heterodimerizations as was reported for G protein-coupled receptors [56].

Figure 3 represents the classification of the allosteric modulators and their allosteric properties: affinity modulation, efficacy modulation, reciprocity, and ceiling effect. Affinity modulation indicates the change in the structural conformation of an orthosteric LBP such that the binding affinity of an orthosteric ligand increases or decreases [57]. Efficacy modulation denotes the increase/decrease in intracellular responses (intrinsic efficacy), depending on the orthosteric ligands (agonist or antagonist) [57]. The ceiling effect or the saturability effect means that allosteric modulators are non-competitive and maintain a certain saturation level at a certain concentration [58]. Allosteric modulators may display the possibility of an absolute subtype selectivity for a target protein. For example, a 1700-fold selective allosteric inhibition for phospholipase D1 (PLD1) compared to its subtype PLD2 has been reported [59]. Allosteric modulators may improve the efficacy and potency of an agonist (e.g., from 2 to 100-fold) in a receptor or may have a synergistic effect [60,61]. Furthermore, reciprocity or allosteric activation may occur [54], which means that the receptor is activated directly, without the presence of an orthosteric ligand [62].

Currently, 202 allosteric modulators have been reported for nuclear receptors [65], with the following existing allosteric ligand binding surfaces: (i) the AF-2 site; (ii) the binding function 3 (BF-3 site); (iii) the ligand-binding pocket (synergistic); (iv) zinc fingers and response elements; and (v) the AF-1 site [66,67]. Allosteric modulators have been demonstrated for numerous nuclear receptors; for example, Gabler et al. reported imatinib as a first-in-class allosteric farnesoid X receptor (FXR) modulator that enhances agonist-induced FXR activation in a reporter gene expression assay. The imatinib analogues-16 (I-16) possess extraordinary efficacy (EC_50_ = 1.9 nm) and high selectivity over other nuclear receptors [68,69].

### 3.1. Allosteric Targeting of the Pregnane X Receptor (PXR)

#### 3.1.1. Duplex—Synergistic Activation of the PXR LBD

The PXR LBD is able to bind two synthetic ligands concomitantly to the orthosteric ligand binding site (Figure 4A) [70], a phenomenon which has been described previously for the PPARγ and ERβ nuclear receptors [71,72]. Two or more compounds accommodate the same binding sites in the receptors and occupy a space within the canonical LBP near the helix 3 region, leading to enhanced coactivator binding, transactivation, and target gene expression [70,71,72].

In the case of the PXR, it was found that binary cocktails of the pesticide *trans*-nonachlor (TNC) as well as 17α-ethinylestradiol (EE2), the active component of contraceptive pills, produce synergistic activation of the PXR and increase expression of its target *CYP3A4* gene. Interestingly, the single chemicals (EE2 or TNC) were not observed to express the PXR target gene *CYP3A4* in the human hepatocyte [70]. Synergistic activation of the PXR by EE2 and TNC reached full agonist activity compared to the potent agonist SR12813, but the compounds act as weak agonists when used separately (Figure 4B).

Small molecules (<500 Da) bind and fill a limited portion of the PXR LBP, leaving empty volume available to accommodate a second compound [73]. This concept has been proposed as a “supramolecular ligand” assembled from two or more compounds that interact with each other within the LBP of a receptor. TNC binds to the LBP, and EE2 binds to the closely adjacent helix 12 in the PXR LBP (Figure 4C). This ectopic site near helix 12 may be considered an allosteric site, with bitopic ligands linking orthosteric and allosteric sites to achieve improved PXR affinity or selectivity [70,74]. The EE2 binding position is a resisted region of the PXR LBP that leaves a significant portion of the pocket unoccupied and available for additional interactions. Supramolecular ligands of the PXR and their properties have been reported to a very limited extent so far, and we can expect many more compounds with this activity [70].

#### 3.1.2. The AF-2 Binding Site at the PXR LBD and Its Ligands

It was found that ligands binding to the AF-2 function of the PXR alter the interactions between a coregulator peptide and the PXR. The ligand-dependent groove of the AF-2 is made up of helixes 3, 4, 5, and 12, which are mainly hydrophobic regions [37]. The orthosteric PXR LBP ends with a short helix 12 (H12), which is important for the structural organization of the AF-2 region to recruit transcriptional coregulators. Coregulators play an important functional role in the transduction of PXR signals. Both corepressors and coactivators bind to the AF-2 regions through a short amphipathic helical sequence containing the Leu-Xxx-Xxx-Leu-Leu (LXXLL) motif in coactivators or Ile/Leu-Xxx-Xxx-Ile/Val-Ile motifs in corepressors via an electrostatic interaction [75,76]. When the binding of a ligand is performed, helix H12 undergoes a significant conformational structural change that alters the overall shape of the AF-2 binding site. The PXR ligands alter the AF-2 site after binding into the PXR LBP, and thus modify the recruitment of coactivators or corepressors, resulting in different agonist or antagonist effects of the ligands [77].

All available PXR antagonist molecules act through similar agonist activation mechanisms, except for some AF-2 disruptors (Figure 5). The PXR’s ligand binding site is the same for both agonists and antagonists, but there are small residue differences with helix 12 [39,40,78]. In addition, none of the identified AF-2 site binding antagonists possess allosteric properties (selectivity, efficacy, or affinity) (Table 1) [54]. Targeting of the AF-2 site always involves issues of nuclear receptor selectivity because the AF-2 site is structurally conserved across subtypes of the nuclear receptor family [78,79]. For example, ketoconazole, the first identified PXR AF-2 site allosteric modulator and antagonist, is a common inhibitor of activated PXRs, CARs, LXRα/βs, and FXRs [80,81]. Ketoconazole binding to the surfaces of the AF-2 site suggests that ketoconazole directly blocks a coactivator (e.g., SRC-1) binding, a finding which was confirmed by a double mutant model (T248E/K277Q) in the AF-2 region of the PXR [81].

The phytoestrogen coumestrol is a natural PXR antagonist proposed to bind to a non-LBP. Biochemical binding assays and LBP-filled mutant (obliterating) studies confirm their surface binding is distinct from the orthosteric site of the LBP [82]. Additionally, computational pharmacophore and docking analyses showed that the known PXR antagonists coumestrol and sulforaphane also accommodate the AF-2 ligand-binding site [83].

In another study, the azole compound FLB-12, a derivative of the azole antifungal ketoconazole, antagonized activated PXRs in the hepatocyte cell line and in in vivo models. The triple mutant plasmid of the PXR LBP (S247W/S208W/C284W) was used to confirm its binding sites outside the PXR LBP. FLB-12 disrupts the interaction between the PXR and SRC-1, a finding which was verified by the protein pull-down assay, indicating the location of binding into the AF-2 sites. This antagonist was found to be selective and less toxic as compared to ketoconazole [84].

Ekins et al. reported residues of the PXR AF-2 ligand binding site. The identified AF-2 ligand binding site is predominantly hydrophobic, and it consists of 15 amino acids (Lys252, Ile255, Lys259, Phe264, Ile269, Glu270, Gln272, Ile273, Ser274, Leu276, Lys277, Pro423, Leu424, Glu427, and Leu428). Lys277 probably serves as a “charge clamp” for the interaction between the coactivator SRC-1 (His687) and the PXR, and it may play a significant role in the initial phase of accommodation of azole molecules into the binding groove of the PXR. Two more azole analogs, enilconazole and fluconazole, have also had interactions confirmed with the AF-2 ligand binding site, as well as antagonist activity shown towards the PXR [80,85].

**Figure 5 cells-11-02974-f005:**
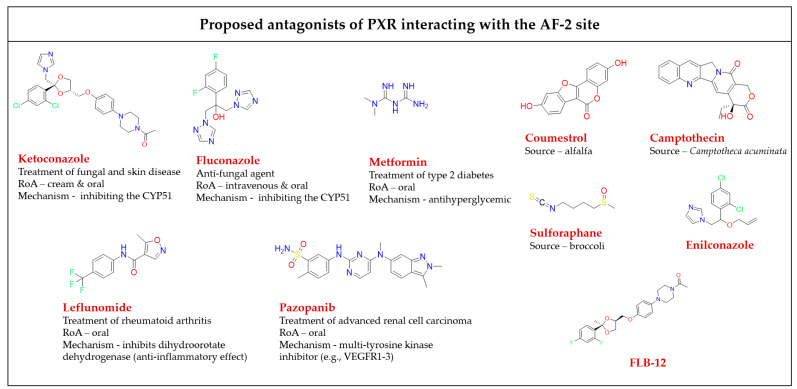
Chemical structures of known AF-2 site-binding PXR antagonists. RoA-routes of administration. The chemical structures were drawn in ChemDraw 20.0 software [80,81,82,83,84,86,87,88,89].

**Table 1 cells-11-02974-t001:** A detailed list of antagonists that bind to the PXR AF-2 binding site.

Molecules	PXR Biological Properties			Binding Site	Reference
	Efficacy (IC_50_)	Affinity (Ki)	NR Selectivity		
Ketoconazole	74.4 µM	55.3 µM	Non-selective	AF-2	[80]
Coumestrol	12 µM	13 µM	Non-selective	AF-2/LBP	[82]
Enilconazole	~20 µM	NA	NA	AF-2	[80,83]
Fluconazole	~20 µM	NA	NA	AF-2	[80,83]
FLB-12	≥23 µM	NA	Selective	AF-2	[84]
Leflunomide	6.8 µM	NA	Non-selective	AF-2	[83,90]
Sulforaphane	12 µM	16 µM	Selective	AF-2	[83,89]
Metformin	NA	>1 mM	Non-selective	AF-2	[87]
Camptothecin	580 nM	NA	Non-selective	AF-2	[88]
Pazopanib	4.1 µM	NA	Selective	AF-2	[86]
Pimecrolimus	1.2 µM	NA	Selective	AF-2/LBP	[86]
73	8.3 µM	NA	Selective	AF-2	[42]

NA-Data not available and NR-Nuclear receptor.

Leflunomide, a drug used clinically for rheumatic arthritis therapy, acts as a PXR antagonist, but as an activator of the CAR [90]. It was shown that it inhibits the PXR/SRC-1 interaction as demonstrated by site-directed mutagenesis of the AF-2 sites [83,91].

The pentacyclic alkaloid camptothecin is known as a topoisomerase I inhibitor, and its analogs are approved for colon cancer therapy. Camptothecin was identified as attenuating *CYP3A4* induction by blocking PXR activation via binding outside of the PXR LBP to prevent the recruitment of coactivators (such as SRC-1) [88].

Metformin is an antihyperglycemic agent used for the treatment of type 2 diabetes mellitus. In two-hybrid assays, we have reported that metformin interrupts the PXR’s interactions with the SRC1 coactivator and it antagonizes PXR-mediated regulation of the *CYP3A4* gene in human hepatocytes. Metformin also inhibits gluconeogenesis by activating AMPK, which is necessary for the transactivation of the PXR. In addition, the compound exhibits a similar effect on the transactivation of other nuclear receptors such as the CAR and VDR [87].

Pazopanib, a tyrosine kinase inhibitor, has recently been identified as a novel potent selective antagonist of PXR activation. The compound was observed to act like camptothecin, a known coactivator disruptor. Pazopanib was claimed as an allosteric noncompetitive antagonist binding in the AF-2 site, which was confirmed with the limited proteolytic digestion technique as well as with the competitive ligand binding TR-FRET PXR coactivator assay. Still, biophysical studies are needed to authorize the positioning of the binding to AF-2 for this antagonist [86].

In addition, the T-cell lymphoma-targeting drug belinostat antagonizes drug-activated PXR-mediated gene expression. Binding assays confirmed belinostat binding to both the LBP and AF-2 binding sites, and molecular docking studies reveal that it is possible to bind to the helix 8 position to allosterically suppress PXR activation [92]. Mustonen et al. discovered a dual PXR and protein kinase inhibitor to prevent PXR-dependent chemoresistance in intestinal carcinoma cells. The two novel analogues of phenylaminobenzosuberone were identified as kinase inhibitors that concomitantly antagonize the PXR. Interestingly, these analogues 100 and 109 are structurally related but functionally different in the PXR. Compound 73 was identified as the mixed competitive and allosteric modulator of the PXR, which was confirmed with the LBP-filled triple mutant model [42].

## 4. Examination of Novel Allosteric Sites for the PXR

### 4.1. The Binding Function 3 (BF-3) as a Novel Allosteric Binding Site for PXR Modulation

The surface BF-3 site on nuclear receptors represents another attractive position for discovering antagonistic molecules to regulate the binding of coactivators [76,93,94,95].

The structural and functional data of the androgen receptor (AR) showed first the presence of another ligand binding site called BF-3, which is located in an area distinct from AF-2 but topographically adjacent. The novel BF-3 site allosterically influences the association of coregulators with AF-2. It was shown that the binding of 3,3‘, 5-triodothyroacetic acid (TRIAC) to BF-3 remodels the adjacent interaction site AF-2 to weaken coactivator binding, as was confirmed by X-ray crystallography. Subsequently, several allosteric inhibitors for the AR were developed, including flufenamic acid (FLUF), triiodothyronine (T3), and some novel compounds (ZINC ID: ZINC12342, ZINC2058890, ZINC3877300, and ZINC3445992) [93,95].

The AR-BF-3 is located on the *N*-terminal helix 1 (residues Gln670, Pro671, Ile672, and Phe673), helix 3 (Pro723, Gly724, Arg726, and Asn727), the loop between helices 3 and 4, and helix 9 (Phe826, Glu829, Leu830, Asn833, Glu837, and Arg840) in the AR (Table 2). The amino acid residues R726 and N727 interlink AF-2 and BF-3 sites, thus transforming allosteric signaling. These detailed structural features to locate BF-3 binding sites are useful for linking this concept to other nuclear receptors. Studies confirmed that the BF-3 position is also available in other nuclear receptors and the helix 3–4 loop (H3 and H4) is thought to be the signature sequence for the BF-3 position (Figure 6) [95,96].

Mutations in the AR-BF-3 site have shown a significant increase in AR activity, indicating that the AR-BF-3 site could be a co-repressor binding site, although this needs to be confirmed. Thus, its most remarkable feature is its interaction with the AF-2 surface conformation and its role in modulating the AF-2 capabilities to engage coactivator peptides [97,98,99]. Katja et al. identified novel GARRPR hexapeptide repeat sequence co-regulator motifs in the AR LBD that allow the binding of Bag-1L co-chaperon peptides. Biochemical assays and molecular modeling studies reveal that the allosteric BF-3 site is an essential domain for the interaction of Bag-1L peptides with the GARRPR motif. The disruption of Bag-1L/AR interactions via allosteric sites or residues in the BF-3 pocket represent targets for the treatment of prostate cancer [100].

The BF-3 ligand binding site, therefore, opens a new paradigm for the development of novel allosteric modulators, as opposed to the targeting of the complex orthosteric site in the PXR [37,38,51,100].

### 4.2. Sequence Homology of the BF-3 Site with Other Nuclear Receptors

The crystal structures of nuclear receptors (PXRs, CARs, and VDRs) have enabled the identification of residues of the BF-3 site. Table 2 lists the BF-3 residues of the PXR that are located in the topologically equivalent position in the solved three-dimensional structure of AR-BF-3 residues (Figure 6). Additionally, we examined putative BF-3 sites of the CAR and VDR. Primary sequence similarity analysis reveals that the typical BF-3 residues are not conserved in other nuclear receptors, in contrast to the PXR (Figure 7A). The AR Q670 is conserved in all the nuclear receptor BF-3 sites. The PXR BF-3 site seems to be unique compared to other analyzed receptors (Table 2). The residue AR P723 is structurally conserved in the VDR (P249), FXR (P310), and CAR (P180), but not in the PXR (S262). Next, we compared structural data reported for the AR and FXR BF-3 residues by multiple sequence analysis using the Clustal Omega tool [46,96,97,100]. Based on our analysis, an alternative BF-3 ligand binding site has been proposed for the PXR (Figure 7B,C).

However, more structural biophysical studies are needed to confirm the location and functionality of the allosteric site. High-throughput screening is required to test compounds in functional or biochemical assays. To accelerate the identification of selective or nonselective BF-3 modulators, new assays and technologies must be developed, since currently available methods only report the detection of a small molecule binding to the orthosteric ligand-binding site of the PXR.

### 4.3. Perspective of Novel PXR Allosteric Binding Sites

Artificial intelligence (AI) is the fastest-growing technology in the life sciences and drug discovery. Computational docking, molecular dynamics simulations, and machine learning approaches are useful for designing and discovering new chemical entities with nuclear receptors [101,102]. Support vector machine algorithms (SVM) and pocket-based analysis have been used to predict allosteric sites in proteins. AI is receiving more attention in the drug discovery field and helping to advance the discovery of new allosteric modulators [103,104]. SVM is used to map and recognize similar data sets through the use of machine learning algorithms [105].

The AlloFinder web server represents a useful tool in new automated drug discovery strategies to identify allosteric modulators [104,106]. The tool was used to identify, e.g., the STAT3 inhibitor K116. Mutational and binding studies further confirmed the inhibition activity of K116 in a novel allosteric site [106]. Subsequently, AlloFinder was used to identify the allosteric site and modulators for the surface antigen CD38. The allosteric modulator LX-102 was found to target CD38 on the side opposite its enzymatic binding pocket, with this confirmed by surface plasmon resonance (SPR) and HDX-MS experiments [107].

Utilizing this web server, we identified a novel allosteric binding pocket for the PXR, which is distinct from the orthosteric LBP, AF-2, and BF-3 regions. The pocket has been termed allosteric site III. In the analysis, the PXR crystal structure was obtained from the protein data bank (PDB) (PDB-5 × 0 R) and the refined structure was submitted to the server to predict allosteric sites and for hotspot mapping (Figure 8).

According to the computational calculation, the allosteric site III is located on helix 3 (α3) and the PXR unique beta sheets β1, and β1′. The identified allosteric binding residues are Thr165, Phe166, Ser167, Phe169, Asn171, Phe172, Leu174, Pro175, Val177, Val211, Leu213, Gln214, Leu215, Arg216, Trp223, Asn224, Tyr225, His242, Cys301, Arg303, Leu304, and Tyr306 (Table 3).

### 4.4. PAM Antagonism: An Emerging Concept in Receptor-Based Drug Discovery

Antagonist molecules can correct inappropriate pathological signaling in two therapeutic settings: (i) prevention of pathological signaling before it is initiated; and (ii) reversal of such signaling once it is established. There are fundamentally two ways in which an antagonist molecule can interact with a receptor to block an agonist response: through an orthosteric or allosteric mechanism. An orthosteric blockade occurs when the antagonist physically binds to the agonist-binding site and prevents agonist binding [63].

The flexible and large LBP of the PXR enables the binding of a wide range of structurally unrelated endogenous and exogenous ligands [33,76]. Preventing pathological signaling before it is initiated by an agonist is difficult for competitive PXR antagonist molecules in the orthosteric PXR LBP. Inhibition is based on association and dissociation rates such as the offset (agonist dissociates from the receptor) and onset (antagonist binds to the receptor) of an agonist and antagonist [63].

It can be supposed that allosteric PXR modulators bind to their distinct sites on the receptor to cause a change in the conformation of the PXR protein that then alters the orthosteric agonist’s effect on the receptor (Figure 2). The new concept of the positive allosteric modulator antagonism for NRs (PAM antagonist) is based on existing GPCR allosteric modulators (e.g., palonosetron for a 5-HT_3_ receptor) [63,108].

PAM antagonists represent a unique class of negative allosteric modulators that antagonize the response of an agonist by increasing the affinity of the agonist to the receptor but decreasing its efficacy, making agonism overall less effective (Figure 9). These divergent reciprocities of the allosteric effect promote a “seek and destroy” mechanism of action [63,109].

PAM antagonism may offer a better way to block pathological receptor signaling than does orthosteric (antagonist) and allosteric (NAM) blockers in order to correct inappropriate pathological signaling in two therapeutic settings: (i) prevention of receptor activation; and (ii) reversal of preexisting agonist activation [63,110].

The development of PAM antagonists is not easy. Functional assays and other emerging technologies such as advanced biophysical and biochemical assays, high-throughput CRISPR engineering and mutant techniques, and molecular docking to X-ray crystal structures could be used for this unique challenge [63]. We can propose that PXR PAM antagonist molecules could be refined to target specific endogenous signaling pathways with special therapeutic effects [63].

**Figure 9 cells-11-02974-f009:**
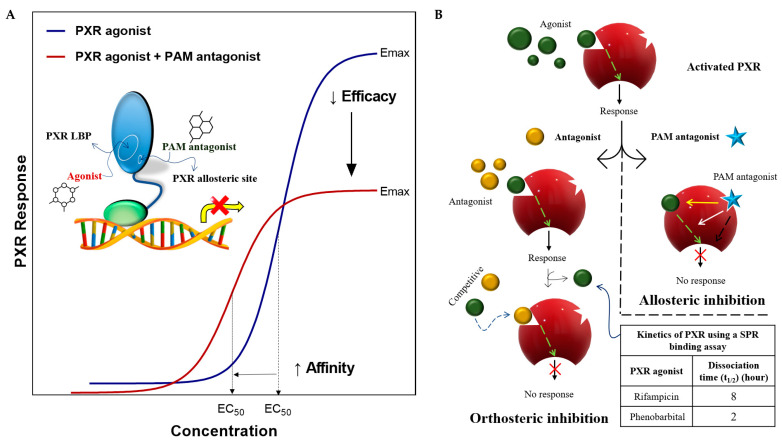
Mode of allosteric modulation by a PAM antagonist. (**A**) A PXR agonist dose-response curve is shown in blue, and the effect of an allosteric modulator (PAM antagonist) on the agonist effect is shown in red. A PAM antagonist decreases efficacy (E_max_) and acts as an antagonist but increases the affinity of the agonist (with lower EC_50_) for site occupancy. Agonist-activated PXRs will decrease the level of gene expression and delay the dissociation of the PXR agonist through the PXR PAM antagonist [63,110]. (**B**) Orthosteric antagonist versus allosteric PAM antagonist receptor mechanisms in an agonist-activated receptor. Orthosteric antagonist: the antagonist (yellow) competes with the agonist (green) to occupy the orthosteric site and/or wait for the dissociation of the agonist from the receptor (e.g., rifampicin dissociation rate (t_1/2_) is 8 h) [111]. Allosteric PAM antagonist: The PAM antagonist directly binds to its distinct site of the receptor and initiates its antagonist function. At the same time, this increases the affinity of the agonist to prevent its dissociation [63]. EC_50_ represents the concentration of a drug that induces a half-maximal response.

## 5. Conclusions

Most ligands bind to the orthosteric PXR ligand binding pocket, which results in transcriptional upregulation (induction) or downregulation (transrepression) of its target genes. Allosteric modulation of the PXR opens up many questions with respect to clinical application and consequences for DDIs. At present, we do not have data for the clinical consequences of PXR inhibition on the putative downregulation of key target PXR genes. Moreover, PXR antagonism has been proposed to alleviate DDIs mediated by PXR inducers [87,112]. Therefore, we are at the beginning of the discovery of efficient PXR allosteric modulators that can help us titrate drug metabolism as well as eliminate PXR-mediated DDIs.

A growing number of small molecules have been shown to bind to the PXR AF-2 coactivator binding site, although none of them have significant affinity (in nanomolar concentration) to the allosteric site (Table 1). The binding site of PXR has not yet been studied in detail because it is challenging to crystallize PXRs without a coactivator. Biochemical assays and technologies must be developed to accelerate the identification of more efficient and highly selective allosteric PXR modulators in the AF-2 site.

In addition, the crystal structure of PXR and molecular modeling have enabled us to identify novel PXR allosteric sites. The novel allosteric ligand binding III and BF-3 sites have been proposed for the PXR in the report, although this awaits confirmation using structural and biophysical techniques. The identified allosteric sites (Figure 10) provide new information in the development of safe and efficient allosteric modulators of the PXR receptor, a promising target for treating chronic metabolic diseases and cancers.

Moreover, other avenues for targeting the PXR are via the development of micro/siRNA strategies (to block or promote the degradation of PXR mRNA), or the design of peptidomimetic inhibitors of the PXR–coactivator interaction. Artificial intelligence combined with advanced chemical-biological and high-throughput screening technologies are expected to help us with the development of novel allosteric modulators of PXRs to fight against metabolic disorders, drug resistance, and colon and breast cancers.

The overview presented in this review may stimulate interest and facilitate scientific effort towards the development of allosteric modulators for the PXR nuclear receptor.

## Figures and Tables

**Figure 1 cells-11-02974-f001:**
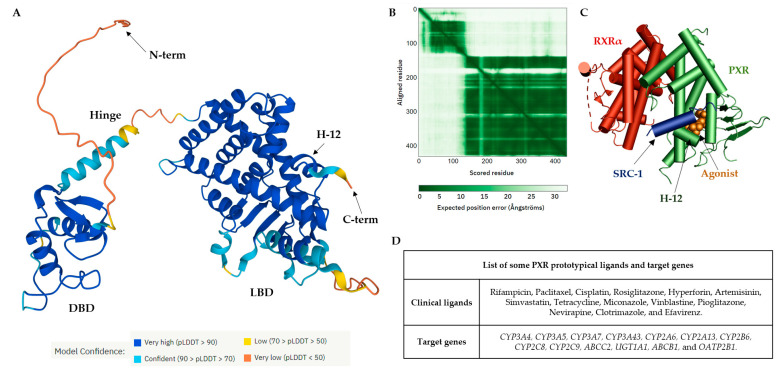
Structure of the PXR receptor. (**A**) AlphaFold2 prediction of the full-length chain of apo-PXR model (UniProt-O75469). A ribbon model of PXR coloring ranges from blue (high confidence of the predicted structure) to red (low confidence of the predicted structure). (**B**) Predicted aligned error plot (PAE) for the PXR. The shade of green indicates expected distance error in angstroms. Dark green represents low error, and light green represents high error indicated in the *N*-terminal/hinge regions of PXR. (**C**) Crystal structure of the PXR/RXRα LBD complex. The liganded PXR (green) and RXRα (red) bind to the steroid receptor coactivator peptide 1 (SRC-1/blue). The PXR agonist SR12813 is indicated in orange (PDB-4J5W) [7]. (**D**) List of some prototypical ligands and target genes for the nuclear receptor PXR.

**Figure 3 cells-11-02974-f003:**
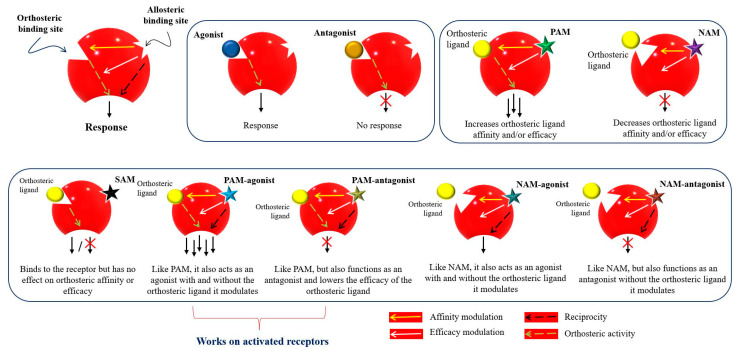
Mode of action and classification of allosteric modulators. Graphical representation of orthosteric and allosteric sites of a receptor. Allosteric ligands bind to a topographically distinct site on a receptor to modulate orthosteric ligand affinity (yellow arrow) and/or efficacy (white arrow). Some allosteric ligands can directly activate their signaling (black arrow). Orthosteric ligands (agonist or antagonist) are colored yellow. Star-shaped allosteric modulators are presented with different colors. PAM—positive allosteric modulator; NAM—negative allosteric modulator; SAM—silent allosteric modulator [53,57,63,64].

**Figure 4 cells-11-02974-f004:**
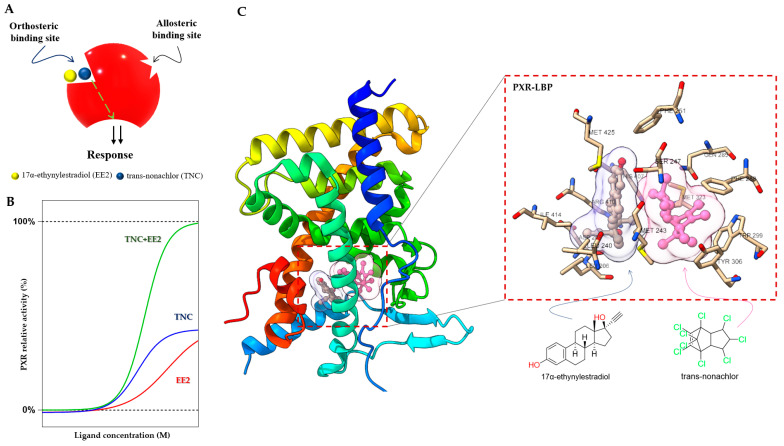
Duplex—synergistic activation of the PXR LBD. (**A**) Graphical representation of the duplex model; two ligands (EE2 and TNC) bind to the orthosteric ligand binding site on a receptor to increase the efficacy. (**B**) The dose response simulation shows that the combination of TNC and EE2 produces synergistic effects on PXR activation. (**C**) Crystal structure of the PXR LBD in complex with EE2 and TNC (PDB-4 × 1 G) and their location highlighted in the ligand binding pocket with interacting amino acid residues [70,73].

**Figure 6 cells-11-02974-f006:**
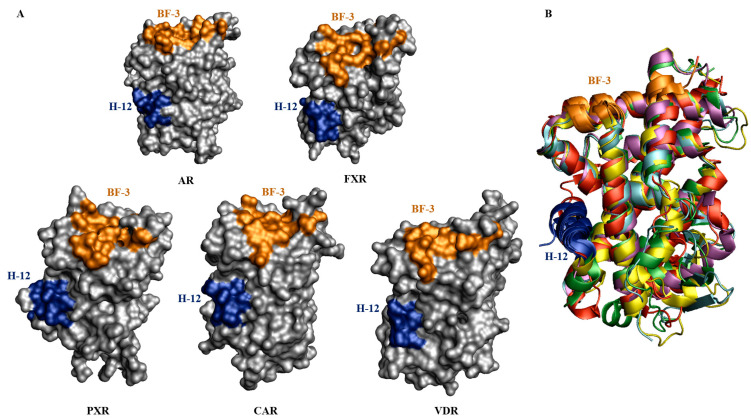
Computational comparison of BF-3 binding sites of selected nuclear receptors by structural superimposition. (**A**) Nuclear receptor surfaces (AR, FXR, PXR, CAR, and VDR) are colored gray, BF-3 allosteric sites are colored orange, and the position of helix 12 (H12) is colored blue. (**B**) Cartoon structure showing superimposed nuclear receptors and their hypothetical BF-3 binding sites (orange) on top of the receptors. The visualized structure is colored: AR-red, FXR-green, PXR-cyan, CAR-yellow, VDR-magenta, and helix 12-blue [96].

**Figure 7 cells-11-02974-f007:**
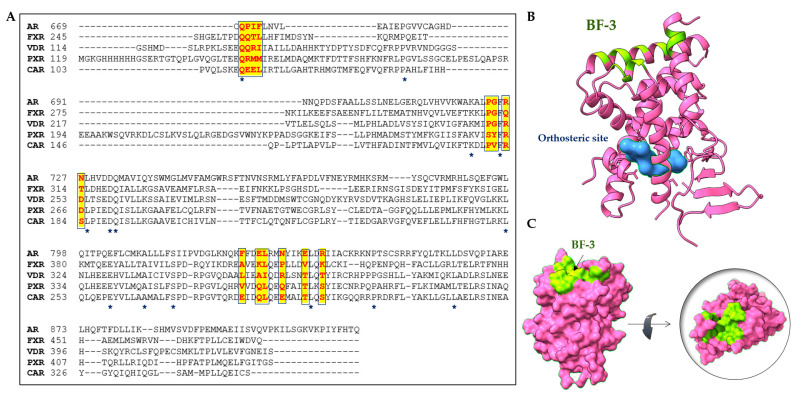
The allosteric binding site BF-3 of the PXR LBD and structural comparisons of the LBD domain between human PXR, CAR, VDR, AR, and FXR receptors. (**A**) Multiple sequence alignment and analysis of PXR, CAR, VDR AR, and FXR receptors show conserved regions. The conserved residues are indicated by the star symbol (*) and the BF-3 residues are highlighted in yellow. This sequence alignment was performed using the Clustal Omega tool (1.2.4) [46,95,97]. (**B**) Crystal structure of the PXR ligand binding domain (pink) showing the ligand binding pocket (blue), and allosteric BF-3 residues are colored green (PDB-5 × 0 R). (**C**) The surface structure of the PXR represents the top view of the BF-3 site (right) and the location of the BF-3 pockets. Structures B and C are in the same orientation.

**Figure 8 cells-11-02974-f008:**
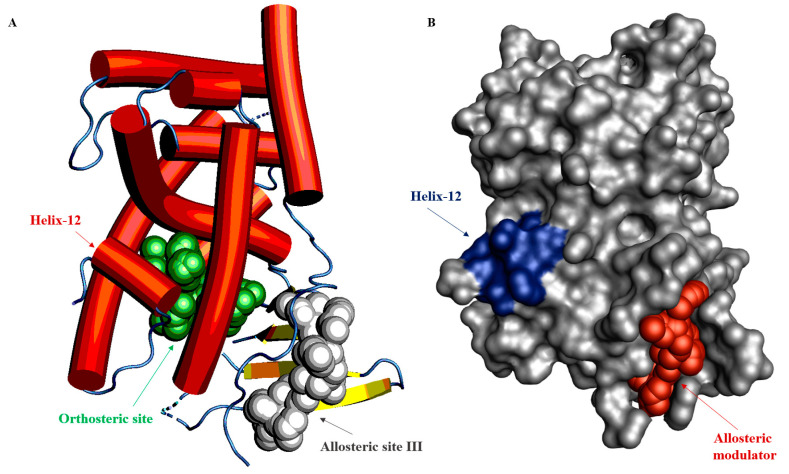
Machine-learning technologies to predict PXR allosteric sites. (**A**) Cartoon structures of the PXR illustrate the orthosteric and allosteric sites. Crystal structure (PDB–5 × 0 R) was used to predict a novel allosteric site using the AlloFinder online server. The green spheres depict the orthosteric ligand binding pocket, and the identified allosteric binding site III is indicated using grey spheres as a ligand. Red arrows point to the helix 12 position. Helix 3 and a beta-sheet (yellow) form the allosteric ligand binding site III. (**B**) Surface structure representation of the allosteric binding site III. The surface of the PXR receptor is gray. Helix 12 is indicated in blue and a hypothetical allosteric modulator in red. Both structures are in the same orientation.

**Figure 10 cells-11-02974-f010:**
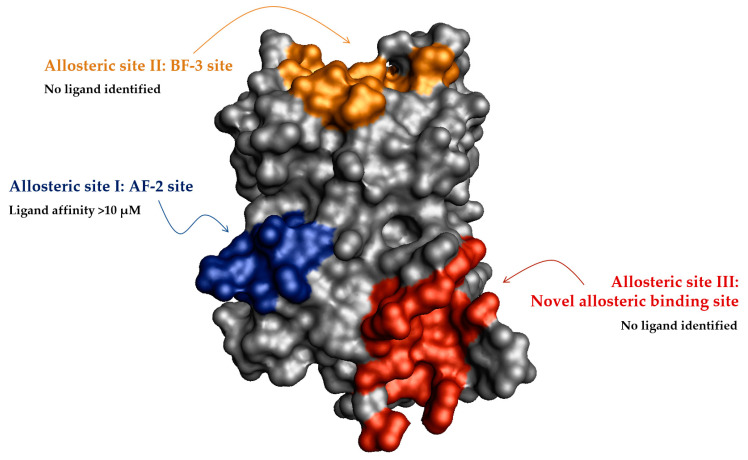
Overview of the surface of the nuclear receptor PXR allosteric sites. Allosteric site I: AF-2 site highlighted in blue; allosteric site II: BF-3 colored orange; allosteric site III in red.

**Table 2 cells-11-02974-t002:** BF-3 residues for PXRs, CARs, and VDRs. The residues were obtained using alignment with the AR and FXR sequences in topologically equivalent positions. ARs and FXRs serve as allosteric site control [96].

Nuclear Receptors	PDB Code	BF-3 Residues													
AR	1T5Z	Q670	P671	I672	F673	P723	G724	R726	N727	F826	E829	L830	N833	E837	R840
FXR	1OSH	Q253	Q254	T255	L256	P310	G311	Q313	T314	A407	K410	L411	P414	V418	K421
VDR	1DB1	Q128	Q129	R130	I131	P249	G250	R252	D253	L351	A354	I355	R358	T362	T365
PXR	1ILH	Q147	R148	M149	M150	S262	Y263	R265	D266	V361	Q364	L365	Q368	T372	S375
CAR	1XVP	Q110	E111	E112	L113	P180	V181	R183	S184	E280	Q283	L284	E287	T291	S294

**Table 3 cells-11-02974-t003:** Predicted residues in the PXR allosteric site III.

Amino Acid Groups	PXR Residues
Hydrophobic	Phe166, Phe169, Phe172, Pro175, Val177, Val211, Leu213, Leu215, Trp223, Leu304
Hydrophilic	Uncharged—Thr165, Ser167, Asn171, Gln214, Asn224, Tyr225, Cys301, Tyr306. Basic—Arg216, His242, Arg303.

## Data Availability

Primary data generated in a publicly available allofinder web server (http://mdl.shsmu.edu.cn/ALF/, accessed on 15 April 2022), and multiple sequence alignment analyzed by the Clustal Omega tool (1.2.4) (https://www.ebi.ac.uk/Tools/msa/clustalo/) (accessed on 15 April 2022).

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
