# Peer review of "Allosteric Antagonism of the Pregnane X Receptor (PXR): Current-State-of-the-Art and Prediction of Novel Allosteric Sites"

_cells, 2022, doi:10.3390/cells11192974_

Round 1
Reviewer 1 Report
Comments on the manuscript entitled „ Allosteric antagonism of the pregnane X receptor (PXR): current-state-of-the-art and prediction of novel allosteric sites”
Kamaraj et al. provide with their review article an excellent and up-to-date article about allosteric antagonism of PXR. Given the great importance of PXR in the pharmacokinetics of drugs (by promoting unwanted DDIs) and (patho)physiology, it can be anticipated that the article will be of interest in the pharmacological community. In addition to summarizing available data, the authors furthermore described the novel binding function 3 (BF-3) site (known from other nuclear receptors) and proposed a novel allosteric site III based on in silico predictions.
However, I am not an expert in in silico modeling of proteins or predicting binding affinity via QSAR or other approaches. Thus, the paper should be in addition carefully reviewed by an respective expert. have only the following comments / questions:
1. The authors state in their Abstract and also in the Introduction that PXR antagonists may have the potential as drug candidate to treat colon cancer. The authors need to cite convincing in vivo studies (animals / human) to prove this statement.
2. In line 42, the authors mentioned that PXR is highly expressed in human intestine and liver. However, of the given citations, only Xang et al. 2020 ( a review article) provides evidence for this. The authors should cite original research papers to prove this statement. Indeed, the expression in terms of protein amount is very little in intestine and liver.
3. The authors mention that ligands (activators) of PXR may cause unwanted DDIs. However, as not all readers are experts for this scenario of interaction, the authors should describe the mechanism in a more detailed manner and may provide some clinical examples.
4. Likewise, the authors should mention prototypical activators of PXR such as rifampin, St. John’s wort and the regulated genes of metabolism and transport. For the latter, I would suggest an additional table comparable to Table 1.
5. Finally, the authors may also discuss in more detail the potential clinical consequences of the described / proposed allosteric antagonists.
6. Are there potential interferences of the antagonists with other nuclear receptors as reported for leflunomide?
Author Response
- The authors state in their Abstract and also in the Introduction that PXR antagonists may have the potential as drug candidate to treat colon cancer. The authors need to cite convincing in vivo studies (animals / human) to prove this statement.
REPLY: We thank the reviewer for these comments. We included relevant research articles with references (26-30) performed with animal and spheroid models in the revised manuscript.
.
References included in the manuscript:
- Wang, Hongwei, et al. "Pregnane X receptor activation induces FGF19-dependent tumor aggressiveness in humans and mice." The Journal of clinical investigation 121.8 (2011): 3220-3232.
- Bansard, Lucile, et al. "Niclosamide induces miR-148a to inhibit PXR and sensitize colon cancer stem cells to chemotherapy." Stem Cell Reports 17.4 (2022): 835-848.
- Planque, Chris, et al. "Pregnane X-receptor promotes stem cell-mediated colon cancer relapse." Oncotarget 7.35 (2016): 56558.
- Raynal, Caroline, et al. "Pregnane X Receptor (PXR) expression in colorectal cancer cells restricts irinotecan chemosensitivity through enhanced SN-38 glucuronidation." Molecular cancer 9.1 (2010): 1-13.
- In line 42, the authors mentioned that PXR is highly expressed in human intestine and liver. However, of the given citations, only Xang et al. 2020 (a review article) provides evidence for this. The authors should cite original research papers to prove this statement. Indeed, the expression in terms of protein amount is very little in intestine and liver.
REPLY: The authors thank the reviewer for pointing out the statement in the manuscript. On the basis of the reviewer’s comments, these original research articles are included in the introduction section and cited (9-16) (Page:2, lines: 54-55).
References included in the manuscript:
- Protein Atlas (https://www.proteinatlas.org/ENSG00000144852-NR1I2/tissue)
- Nishimura, Masuhiro, Shinsaku Naito, and Tsuyoshi Yokoi. "Tissue-specific mRNA expression profiles of human nuclear receptor subfamilies." Drug metabolism and pharmacokinetics 19.2 (2004): 135-149.
- Lehmann, Jürgen M., et al. "The human orphan nuclear receptor PXR is activated by compounds that regulate CYP3A4 gene expression and cause drug interactions." The Journal of clinical investigation 102.5 (1998): 1016-1023.
- Kliewer, Steven A., et al. "An orphan nuclear receptor activated by pregnanes defines a novel steroid signaling pathway." Cell 92.1 (1998): 73-82.
- Fukuen, Shuichi, et al. "Identification of the novel splicing variants for the hPXR in human livers." Biochemical and biophysical research communications 298.3 (2002): 433-438.
- The authors mention that ligands (activators) of PXR may cause unwanted DDIs. However, as not all readers are experts for this scenario of interaction, the authors should describe the mechanism in a more detailed manner and may provide some clinical examples.
REPLY: We thank the reviewer for the valuable comment. The following paragraphs have been included in the Introduction section (Page: 2, lines: 58 - 64).
We modified the sentence as follows: “Activation of PXR causes significant clinically relevant DDIs with compounds whose clearance is critically dependent on hepatic biotransformation by inducible cytochrome P450s, such as CYP3A4 (Figure 1D). Under this scenario, the metabolism of the drug is significantly augmented by a PXR ligand (inducer, perpetrator), resulting in a decrease in therapeutic efficacy, which may subsequently require shortening dosage intervals or increasing dosage. The most serious DDIs interactions mediated by PXR activation have been reported for rifampicin [21-24]”.
References included in the manuscript:
- Sinz, Michael W. "Evaluation of pregnane X receptor (PXR)-mediated CYP3A4 drug-drug interactions in drug development." Drug metabolism reviews 45.1 (2013): 3-14.
- Shukla, Sunita J., et al. "Identification of clinically used drugs that activate pregnane X receptors." Drug Metabolism and Disposition 39.1 (2011): 151-159.
- Ma, Xiaochao, et al. "Rifaximin is a gut-specific human pregnane X receptor activator." Journal of Pharmacology and Experimental Therapeutics 322.1 (2007): 391-398.
- Wang, Yue-Ming, et al. "Role of CAR and PXR in xenobiotic sensing and metabolism." Expert opinion on drug metabolism & toxicology 8.7 (2012): 803-817.
- Likewise, the authors should mention prototypical activators of PXR such as rifampin, St. John’s wort and the regulated genes of metabolism and transport. For the latter, I would suggest an additional table comparable to Table 1.
REPLY: Thank you for the comment. We added the most important PXR ligands used in the clinic as well as in experimental conditions into revised Figure 1 (1D). In addition, we mention the most important target genes involved in drug metabolism and clearance in the revised manuscript (Page:2, lines: 46).
FIGURE 1D
- Finally, the authors may also discuss in more detail the potential clinical consequences of the described / proposed allosteric antagonists.
REPLY: We greatly appreciate the reviewers' comments. Based on the comments, we include the following paragraphs in the conclusion section (Page: 14, lines: 456-462).
We modified the sentence as follows: “Allosteric modulation of PXR opens up many questions with respect to clinical application and consequences for DDIs. At present, we do not have data about clinical consequences of PXR inhibition on the putative down-regulation of key target PXR genes. Moreover, PXR antagonism has been proposed to alleviate DDIs mediated by PXR inducers [94,120,121]. Therefore, we are at the beginning of the discovery of efficient PXR allosteric modulators that can help us titrate drug metabolism as well as eliminate PXR-mediated DDIs”.
References included in the manuscript:
- Chen, Yakun, et al. "Camptothecin attenuates cytochrome P450 3A4 induction by blocking the activation of human pregnane X receptor." Journal of Pharmacology and Experimental Therapeutics 334.3 (2010): 999-1008.
- Sinz, Michael, et al. "Evaluation of 170 xenobiotics as transactivators of human pregnane X receptor (hPXR) and correlation to known CYP3A4 drug interactions." Current drug metabolism 7.4 (2006): 375-388.
- Burk, Oliver, et al. "Identification of approved drugs as potent inhibitors of pregnane X receptor activation with differential receptor interaction profiles." Archives of Toxicology 92.4 (2018): 1435-1451.
- Are there potential interferences of the antagonists with other nuclear receptors as reported for leflunomide?
REPLY: The author thanks the reviewer for this comment. Please refer to Table 1, we detail the nuclear receptor selectivity.
Sincerely,
Rajamanikkam Kamaraj and Petr Pávek
on behalf of all authors
Reviewer 2 Report
This paper describes a study on the pregnane X receptor (PXR), a nuclear receptor. The authors perform a detailed analysis of previous work in the allosteric modulation of PXR. They identify a site on PXR homologous to the BF-3 site found on the androgen receptor and other nuclear receptors. Using an artificial intelligence tool, they identify an additional potential allosteric regulatory site on PXR, which they call site III. Then, site III was used for virtual drug screening to identify potential novel allosteric modulators for PXR.
This is an excellent paper that may be published with only a few additions to the text:
In the discussion of the BF-3 site in section 4.2, the authors need to discuss the results of Jehle et al (2014), Coregulator Control of Androgen Receptor Action by a Novel Nuclear Receptor-binding Motif, which showed that the AR BF-3 is a binding site for the Bag-1L protein.
In the discussion of predicting novel PXR allosteric binding sites in section 4.3, the authors should also cite the work of Xu et al (2013), Detection of persistent organic pollutants binding modes with androgen receptor ligand binding domain by docking and molecular dynamics.
Author Response
This paper describes a study on the pregnane X receptor (PXR), a nuclear receptor. The authors perform a detailed analysis of previous work in the allosteric modulation of PXR. They identify a site on PXR homologous to the BF-3 site found on the androgen receptor and other nuclear receptors. Using an artificial intelligence tool, they identify an additional potential allosteric regulatory site on PXR, which they call site III. Then, site III was used for virtual drug screening to identify potential novel allosteric modulators for PXR.
This is an excellent paper that may be published with only a few additions to the text:
In the discussion of the BF-3 site in section 4.2, the authors need to discuss the results of Jehle et al (2014), Coregulator Control of Androgen Receptor Action by a Novel Nuclear Receptor-binding Motif, which showed that the AR BF-3 is a binding site for the Bag-1L protein.
REPLY: Thanks for the comments and suggestions. We include and/or discuss the following details in the mentioned research article on page 9, lines 323 – 328.
“Katja et al. identified novel GARRPR hexapeptide repeat sequence co-regulator motifs in AR-LBD that allow binding of the Bag-1L co-chaperon peptides. Biochemical assays and molecular modeling studies reveal that the allosteric BF-3 site is an essential domain for the interaction of Bag-1L peptides with the GARRPR motif. The disruption of Bag-1L/AR interactions via allosteric sites or residues in the BF-3 pocket represent targets for prostate cancer treatment [101]”.
References included in the manuscript:
- Jehle, Katja, et al. "Coregulator control of androgen receptor action by a novel nuclear receptor-binding motif." Journal of Biological Chemistry 289.13 (2014): 8839-8851.
In the discussion of predicting novel PXR allosteric binding sites in section 4.3, the authors should also cite the work of Xu et al (2013), Detection of persistent organic pollutants binding modes with androgen receptor ligand binding domain by docking and molecular dynamics.
REPLY:Thanks to the reviewer, we included the mentioned research article in the PXR allosteric binding site section on page 11, lines 372 – 374.
We included the sentence as follows: “Computational docking, molecular dynamics simulations, and machine learning approaches are useful for designing and discovering new chemical entities with nuclear receptors [102, 103]”.
References included in the manuscript:
- Xu, Xian Jin, et al. "Detection of persistent organic pollutants binding modes with androgen receptor ligand binding domain by docking and molecular dynamics." BMC structural biology 13.1 (2013): 1-9.
Sincerely,
Rajamanikkam Kamaraj and Petr Pávek
on behalf of all authors
Reviewer 3 Report
Pavek et al. present a review of PXR-targeted allosteric antagonism with a flavor of novel allosteric site prediction. The topic of the review is of high-interest and well excecuted, but the manuscript part related to the in silico prediction of the novel site requires major modifications. Overall, I highly recommend such a valuable manuscript to be accepted if the key-issues mentioned here are properly addressed. Below I list some of the key issues related to the manuscript.
Major issues:
1) Authors display an AlphaFold2 predicted structure in Figure 1. In the image, model confidence is not indicated. This needs to be included into the figure that the readers are not misled while seeing the structure in the image. With AlphaFold predicted structures this needs to be always presented! Moreover, the presented structure seems to be lacking part of the N-terminal.
2) In figure 3 allosteric modulation is described. However, in 3.1.1 “duplex” is described that is not included into the figure. Please modify the figure 3 to also include this.
3) BF-3 region is described as a novel allosteric binding site. The information that is missing in the manuscript is how well this site is conserved within all published PXR structures, is there any conformational changes or how rigid that is?
4) Please describe the druggability of this site with an in silico evaluation (binding site volume, characteristics, some druggability scores etc.).
5) Section 4.3 prediction of allosteric site III, has the following key-issues:
a. The pocket prediction needs to be repeated with multiple PXR structures (now only one is included). How much has this impact on the results?
b. The proposed region includes / is adjacent to loop regions. How stabile are these loops, what kind of effect on the site their fluctuation will induce? Some loop regions are considerably dynamic in PXR based on MD simulations (https://doi.org/10.1016/j.csbj.2022.06.020) and crystal structures.
c. It lacks the information “how good is this allosteric site”? (Refer to the point 4 above).
6) The proposed novel ligands for the allosteric site III are highly speculative at this point, thus cannot be included without any actual experimental data here in my opinion.
Minor points:
7) Line 72 “(>1600 Å); is this referring to the volume Å3 or what?
8) Line 98, the word “binding” can be removed from “hydrogen bond binding”, to avoid repetition and will make text clearer.
9) Line 172 onwards there are listed the PXR allosteric sites; please include an illustration at this point for these sites to make the text more comprehensible to the readers.
10) Line 204 “addresses the issues of selectivity”. Generally, selectivity is quite broad term; therefore, you will need to clearly specify what kind of selectivity is meant in here. Subtype, family, NR selectivity, off-target etc.?
11) Table 1. As the point above; define clearly what is meant with the used terms.
12) Figure 5. A SI image of the close up surfaces with their properties (charge; lipophilicity) would be useful.
13) Table 2. Add some color to highlight the types of the amino acids to improve the table readability (now it is poor).
14) Line 377. “virtual drug screening”; you have screened ligands, not drugs; therefore change the term drug to ligand (or omit the word drug).
15) Line 400: “The inhibition is based on association and dissociation rates such as offset (agonist dissociates from the receptor) and onset (antagonist binds to the receptor) of an agonist and antagonist [55]”. How does the target residence time concept apply with PXR (see RA Copeland papers)? Would long target residence time be decisive for the biological action with PXR ligands?
16) Is gene specific modulation of PXR signaling expected with allosteric agents?
Author Response
Pavek et al. present a review of PXR-targeted allosteric antagonism with a flavor of novel allosteric site prediction. The topic of the review is of high-interest and well excecuted, but the manuscript part related to the in silico prediction of the novel site requires major modifications. Overall, I highly recommend such a valuable manuscript to be accepted if the key-issues mentioned here are properly addressed. Below I list some of the key issues related to the manuscript.
REPLY: We are pleased to receive the following valuable comments from reviewer 3. Each major or minor point strongly improves the manuscript and encourages us to submit the original research article.
Major issues:
1) Authors display an AlphaFold2 predicted structure in Figure 1. In the image, model confidence is not indicated. This needs to be included into the figure that the readers are not misled while seeing the structure in the image. With AlphaFold predicted structures this needs to be always presented! Moreover, the presented structure seems to be lacking part of the N-terminal.
REPLY: We thank the reviewer for this comment to improve Figure 1. Based on the comments, we modified Figure 1 and included the error plot, as well as the confidence of the model from the appropriate source (Figure 1A – B). Terminals C and N are indicated in the arrow marks in Figure 1 A (line 46). We discuss this in the Introduction Section on page 1, lines 41 – 44.
References included in the manuscript:
- Jumper, John, et al. "Highly accurate protein structure prediction with AlphaFold." Nature 596.7873 (2021): 583-589.
- Perrakis, Anastassis, and Titia K. Sixma. “AI revolutions in biology: The joys and perils of AlphaFold.” EMBO reports 22.11 (2021): e54046.
SEE MODIFIED FIGURE 1 IN UPLOADED FILE
2) In figure 3 allosteric modulation is described. However, in 3.1.1 “duplex” is described that is not included into the figure. Please modify the figure 3 to also include this.
REPLY: We thank the reviewer for the comment. We have not modified Figure 3 as suggested, instead, we included the new Figure 4 to visualize duplex allosteric modulation. We also improved text on page 5, lines 189 – 204.
References included in the manuscript:
- Delfosse, Vanessa, et al. "Synergistic activation of human pregnane X receptor by binary cocktails of pharmaceutical and environmental compounds." Nature communications 6.1 (2015): 1-10.
- Delfosse, Vanessa, et al. "Mechanistic insights into the synergistic activation of the RXR–PXR heterodimer by endocrine disruptor mixtures." Proceedings of the National Academy of Sciences 118.1 (2021): e2020551118.
SEE MODIFIED FIGURE 4 IN UPLOADED FILE
3) BF-3 region is described as a novel allosteric binding site. The information that is missing in the manuscript is how well this site is conserved within all published PXR structures, is there any conformational changes or how rigid that is?
REPLY: The authors thank again for the valuable comment that encouraged us to perform pairwise structure alignment.
We performed rigid body alignment in PXR crystal structures using Java-based chaining aligned fragment pairs allowing twists (jFATCAT-rigid). Sequence alignment superimposed results presented here (n=10), RMSD 0.77 Å and target structure coverage 98% observed. All available 49 crystal structure superimposed data and structural sequence-analyzed data will be published in the subsequent invited original research article, as suggested by editors.
Additionally, we have data for a deep learning approach to predict protein flexibility from direct sequences. We found that the predicted BF-3 sites are rigid.
SEE FIGURE IN UPLOADED FILE
4) Please describe the druggability of this site with an in silico evaluation (binding site volume, characteristics, some druggability scores etc.).
REPLY: Thank you for your comments. The volume of the BF-3 allosteric site is ~600 Å3.
In silico analysis and related data will be published in a follow-up invited original research manuscript in the same journal.
5) Section 4.3 prediction of allosteric site III, has the following key-issues:
- The pocket prediction needs to be repeated with multiple PXR structures (now only one is included). How much has this impact on the results?
REPLY: Thank you for the comments.
We can briefly answer your question. We are working on this to accomplish the task with all reported crystal structures using machine learning approaches other than the webserver application. Data we have done with multiple crystal structures we will present later in the focused invited manuscript in the PXR topical collections in Cells. Moreover, here we include our successful result based on the allofinder webserver, pocket prediction repeated with 2QNV (Crystal Structure of the Pregnane X Receptor bound to Colupulone). Unfortunately, based on the recommendation of editors, a detailed description and methodology for allosteric site III characterization will be published in a follow-up original research manuscript in the same journal.
SEE FIGURE IN UPLOADED FILE
However, we strongly believe that this pocket is available for ligand docking. Various data sources (some of which we include here) supported that this site is one of the pocket sites for ligands. Please refer to the following example, which we used to confirm this pocket with BioBB (BioExcel Building Blocks), DoGSiteScorer and PocketDrug (Pocket draggability prediction). Moreover, allosteric activity needs to be confirmed with biological assays.
SEE FIGURE IN UPLOADED FILE
References:
- Bayarri, Genís, et al. "BioExcel Building Blocks Workflows (BioBB-Wfs), an integrated web-based platform for biomolecular simulations." Nucleic Acids Research 50.W1 (2022): W99-W107.
- Hussein, Hiba Abi, et al. "PockDrug-Server: a new web server for predicting pocket druggability on holo and apo proteins." Nucleic acids research 43.W1 (2015): W436-W442.
- Volkamer, Andrea, et al. "DoGSiteScorer: a web server for automatic binding site prediction, analysis and druggability assessment." Bioinformatics 28.15 (2012): 2074-2075.
- The proposed region includes / is adjacent to loop regions. How stabile are these loops, what kind of effect on the site their fluctuation will induce? Some loop regions are considerably dynamic in PXR based on MD simulations (https://doi.org/10.1016/j.csbj.2022.06.020) and crystal structures.
REPLY: Thank you so much for these comments.
We highly appreciate the reviewer pointing out the recent PXR microsecond timescale molecular dynamics research articles. At this time, we do not perform any molecular dynamic simulations studies to confirm the structural conformation of this site and stability to ligand binding.
- It lacks the information “how good is this allosteric site”? (Refer to the point 4 above).
REPLY: Thank you for the comments.
We can briefly answer your question. The identified allosteric site – III druggability properties (scores for hydrophobicity, polarity, aromatic, physicochemical, and geometric (binding site surface volumes) and allosteric scores were evaluated using in silico tools (AlloFinder, PocketDrug and DoGSiteScorer servers) in our ab. Note these data we do not include in the manuscript.
Allosteric Site Score: 0.572 (scale: 0 – low to 1 – high)
Volume(Å3): ~657
Drug-like Score : 0.89 (scale: 0 – low to 1 – high)
Druggability Probability: 0.79 (scale: 0 – low to 1 – high)
Polar Residues Proportion: 0.53 (scale: 0 – low to 1 – high)
Aromatic Residues Proportion: 0.2 (scale: 0 – low to 1 – high)
Hydrophobicity ratio: 0.64
Hydrophobic interactions: 7
Hydrogen bond acceptors: 4
Unfortunately, based on recommendation of editors, a detailed description and methodology for allosteric site - III characterization will be published in a follow-up original research manuscript in the same journal.
References:
- Li, Shuai, et al. "Alloscore: a method for predicting allosteric ligand–protein interactions." Bioinformatics 32.10 (2016): 1574-1576.
- Huang, Min, et al. "AlloFinder: a strategy for allosteric modulator discovery and allosterome analyses." Nucleic acids research 46.W1 (2018): W451-W458.
- Hussein, Hiba Abi, et al. "PockDrug-Server: a new web server for predicting pocket druggability on holo and apo proteins." Nucleic acids research 43.W1 (2015): W436-W442.
- Volkamer, Andrea, et al. "DoGSiteScorer: a web server for automatic binding site prediction, analysis and druggability assessment." Bioinformatics 28.15 (2012): 2074-2075.
6) The proposed novel ligands for the allosteric site III are highly speculative at this point, thus cannot be included without any actual experimental data here in my opinion.
REPLY: Thanks to the reviewer for this opinion. We agree. Considering these comments and editors’ recommendations, we excluded the original data from the manuscript, as well as their related figures 8 and Table 4. The data will be corroborated and published in the future original article.
Minor points:
7) Line 72 “(>1600 Å); is this referring to the volume Å3 or what?
REPLY: Thanks for noticing this error. We changed/revised according to the valuable suggestions of the reviewer on page 3, line 85.
8) Line 98, the word “binding” can be removed from “hydrogen bond binding”, to avoid repetition and will make text clearer.
REPLY: Thank you, we removed unwanted/repeated word binding.
9) Line 172 onwards there are listed the PXR allosteric sites; please include an illustration at this point for these sites to make the text more comprehensible to the readers.
REPLY: We are very happy to get these comments from the reviewer! We include the new figure 4 (line 194), this figure we hope will be useful to readers, and it improves the manuscript to answer the major correction of Comment 2. We also improved the section The duplex allosteric section (3.1.1) on page 5, lines 189 – 204.
10) Line 204 “addresses the issues of selectivity”. Generally, selectivity is quite broad term; therefore, you will need to clearly specify what kind of selectivity is meant in here. Subtype, family, NR selectivity, off-target etc.?
REPLY: Based on the reviewer’s comments, we changed and clearly mentioned “nuclear receptor selectivity” on page 7, line 231.
11) Table 1. As the point above; define clearly what is meant with the used terms.
REPLY: We clearly defined NR selectivity in Table 1 (nuclear receptor selectivity) on page 8, line 296.
12) Figure 5. A SI image of the close up surfaces with their properties (charge; lipophilicity) would be useful.
REPLY: We thank the reviewer for the comment. Unfortunately, based on the recommendation of editors, a detailed description and methodology for allosteric site characterization will be published in a follow-up original research manuscript in the same journal.
We can briefly answer your question. The following figures are close-up surfaces of PXR BF-3 sites showing charge and lipophilicity properties.
SEE FIGURE IN UPLOADED FILE
13) Table 2. Add some color to highlight the types of the amino acids to improve the table readability (now it is poor).
REPLY: Thank you for the suggestion to create a colorful table for better readability. We highlighted different colors based on amino acid residues.
SEE MODIFIED TABLE 2
14) Line 377. “virtual drug screening”; you have screened ligands, not drugs; therefore change the term drug to ligand (or omit the word drug).
REPLY: We thank the reviewer for this comment. We removed this paragraph based on reviewer comments and editors' recommendations.
15) Line 400: “The inhibition is based on association and dissociation rates such as offset (agonist dissociates from the receptor) and onset (antagonist binds to the receptor) of an agonist and antagonist [55]”. How does the target residence time concept apply with PXR (see RA Copeland papers)? Would long target residence time be decisive for the biological action with PXR ligands?
REPLY: We thank the reviewer for these comments.
“The inhibition is based on association and dissociation rates such as offset (agonist dissociates from the receptor) and onset (antagonist binds to the receptor) of an agonist and antagonist [55]”. For this sentence we explain simple dissociation concept (The simple one-step binding and one-step dissociation mechanism/Kd = koff/kon).
The described PAM-antagonist concept is linked with NAM. Residence time model extends the duration of drug action (efficacy/pharmacological activity). In the concept of PAM-antagonist, it blocks the activation and negatively regulates the receptor.
PXR is significantly expressed in pathophysiological conditions (e.g., cancer). In the disease state, the PAM-antagonist blocks activation and negatively regulate the receptor.
References:
- Copeland, Robert A. "Evolution of the drug-target residence time model." Expert Opinion on Drug Discovery 16.12 (2021): 1441-1451.
- Kenakin, Terry, and Ryan T. Strachan. "PAM-antagonists: a better way to block pathological receptor signaling?." Trends in pharmacological sciences 39.8 (2018): 748-765.
- Copeland, Robert A., David L. Pompliano, and Thomas D. Meek. "Drug–target residence time and its implications for lead optimization." Nature reviews Drug discovery 5.9 (2006): 730-739.
16) Is gene specific modulation of PXR signaling expected with allosteric agents?
REPLY: Yes. We believe that allosteric PXR modulators will control or modify specific genes associated with PXR. However, this is only our speculation, and we need data to confirm the idea.
Sincerely,
Rajamanikkam Kamaraj and Petr Pávek
on behalf of all authors
Round 2
Reviewer 3 Report
The authors have made a good work with their revision, and with moving their allosteric ligand discovery to a separate manuscript they have addressed my earlier concerns.
I have just one new point to comment. In new Fig 10, allosteric site III is differently represented. Please use the same style as for other sites to avoid confusion (PXR surface highlight, not a hypothetical ligand).
Author Response
Dear reviewer,
We thank you for the suggestion. According to your comment, Figure 10 has been modified.
Sincerely,
Petr Pavek
